# Acquired Resistance to Antibody-Drug Conjugates

**DOI:** 10.3390/cancers11030394

**Published:** 2019-03-20

**Authors:** Denis M. Collins, Birgit Bossenmaier, Gwendlyn Kollmorgen, Gerhard Niederfellner

**Affiliations:** 1National Institute for Cellular Biotechnology, Dublin City University, Glasnevin, Dublin 9, Ireland; denis.collins@dcu.ie; 2Pieris Pharmaceuticals GmbH, 85354 Freising, Germany; bossenmaier@pieris.com; 3Roche Diagnostics GmbH, 82377 Penzberg, Germany; gwendlyn.kollmorgen@roche.com; 4Beoro Therapeutics GmbH, 82229 Seefeld, Germany

**Keywords:** antibody-drug conjugates, targeted delivery, drug resistance, multidrug resistance proteins, apoptosis resistance, immunotoxins

## Abstract

Antibody-drug conjugates (ADCs) combine the tumor selectivity of antibodies with the potency of cytotoxic small molecules thereby constituting antibody-mediated chemotherapy. As this inherently limits the adverse effects of the chemotherapeutic, such approaches are heavily pursued by pharma and biotech companies and have resulted in four FDA (Food and Drug Administration)-approved ADCs. However, as with other cancer therapies, durable responses are limited by the fact that under cell stress exerted by these drugs, tumors can acquire mechanisms of escape. Resistance can develop against the antibody component of ADCs by down-regulation/mutation of the targeted cell surface antigen or against payload toxicity by up-regulation of drug efflux transporters. Unique resistance mechanisms specific for the mode of action of ADCs have also emerged, like altered internalization or cell surface recycling of the targeted tumor antigen, changes in the intracellular routing or processing of ADCs, and impaired release of the toxic payload into the cytosol. These evasive changes are tailored to the specific nature and interplay of the three ADC constituents: the antibody, the linker, and the payload. Hence, they do not necessarily endow broad resistance to ADC therapy. This review summarizes preclinical and clinical findings that shed light on the mechanisms of acquired resistance to ADC therapies.

## 1. Background and Introduction

The basic principle of antibody-drug conjugates (ADCs) is to enhance the tumor selectivity of cancer treatment with highly cytotoxic small molecules by covalently linking them to antibody molecules directed against tumor-specific cell surface antigens. Although it sounds simple, this underlying principle has proven very difficult to implement in practice, as illustrated by the fact that two decades of ADC development efforts by biotech and pharma companies have so far only yielded four FDA (Food and Drug Administration)-approved, commercially available products. Currently, more than 150 ADC programs are being actively pursued in different stages of preclinical and clinical development [1]. Although few, the success stories prove that a therapeutic window can be achieved with ADCs despite the required use of payloads that are 100–1000 fold more cytotoxic than standard chemotherapeutics. Such extremely high cytotoxic potency of the payloads is necessary because only a very small portion of the applied dose of an ADC is actually delivered to the targeted tumor (depending on the antibody used between 0.003%–0.08% of injected dose/g tumor), while the vast remainder distributes throughout the body, where it is antigen-independently taken up and catabolized by non-target cells. Therefore, it is not too surprising that clinical dosing is limited by adverse effects that are typically off-target and are determined primarily by the combined effect of the linker/conjugation chemistry and the payload used.

ADC linkers are either non-cleavable or cleavable. In the latter case, depending on the chemical nature of the linker, either low pH or lysosomal enzymes or reducing conditions promote cleavage, which allows the payload to escape from endosomes or lysosomes to the cytosol and to exert its cytotoxic effect. In the case of non-cleavable linkers, the antibody has to be completely degraded in lysosomes to allow cytosolic escape of the active metabolite which consists of the payload still attached by the linker to the single amino acid residue it has been conjugated to. For most ADCs, the payload is conjugated via maleimide or succinimide ester derivatization to ε-amino groups of lysine residues or to free thiol-groups of reduced cystine disulfides naturally occurring in the targeting antibody. However, in recent years, site-specific conjugation has been achieved by recombinant substitution of certain antibody residues with cysteines or non-natural amino acids, like *p*-acetylphenylalanine or selenocysteine. Site-specific conjugation results in a more uniform drug-antibody ratio, and for certain conjugation sites, it produces better drug-linker stability leading to improved overall pharmacokinetic properties, efficacy, and safety of the ADC.

The payloads used for ADCs comprise two broad classes of agents: microtubule-disrupting and DNA-damaging agents. The former class is represented by auristatin analogs, maytansinoids, and tubulysins, all of which inherently only kill rapidly dividing cells by inducing mitotic arrest. While this property limits off-target toxicities in non-dividing normal cells, it also allows tumor stem cells to escape from treatment effects. By contrast, DNA-damaging agents represented by calicheamicins, duocarmycins, and pyrrolo-benzodiazepines (PBDs) bind to the minor groove of DNA and exert their mode of action independent of the cell cycle. While they all also kill tumor stem cells, their molecular modes of action differ. Calicheamicins, originally isolated from *Actinomyces* bacteria, induce site-specific DNA double-strand breaks. Duocarmycins, first isolated from *Streptomyces* bacteria, disrupt DNA architecture by irreversibly alkylating the nucleobase adenine at the N3 position. PBDs, also naturally occurring in *Actinomyces* bacteria, are DNA alkylating compounds that site-specifically cross-link DNA without distorting its double helix structure. Besides these two broad classes of payload agents, a limited number of ADC programs alternatively use analogs of the topoisomerase 1 inhibitor, camptothecin, or the RNA polymerase II inhibitor, α-amanitin, as cytotoxic payloads. Both these agents are also claimed to be effective against cancer stem cells [2,3].

Despite employing extremely potent toxic agents, many of which can also kill non-proliferating cancer stem cells, ADCs have not achieved cures in cancer therapy. As is the case with standard chemotherapies, activation of numerous cell stress pathways and selective pressure for mutations that provide relief allow tumor cells over time to also acquire resistance to ADC treatment. However, since an ADC molecule is more complex than just the sum of its targeting and effector parts, mechanisms of tumor escape from ADC treatment have turned out to be more varied than the simple loss of the targeted surface antigen and/or acquired cellular resistance to the payload used. This review summarizes the current knowledge about how tumor cells acquire resistance to ADC therapy by reviewing the clinical experiences with FDA-approved ADCs and the mechanisms of resistance that have been reported for these agents from preclinical or clinical studies. In the final section, we discuss challenges and available options for overcoming the emergence of acquired resistance to ADC therapy.

## 2. FDA-Approved ADCs

Three of the four FDA-approved ADCs are treatments for hematological malignancies, and so far, only one is for a solid tumor indication. The two main reasons for better success in developing ADCs for blood cancers are that tumor selectivity of the targeted antigens is less of a hurdle for blood cancers and that the targeted cancer cells are more accessible than in solid tumors. Table 1 summarizes resistance mechanisms directed against the different components of these ADCs that have been observed in preclinical and/or clinical studies.

### 2.1. Gemtuzumab Ozogamicin

Gemtuzumab ozogamicin (GO; Mylotarg™) is composed of a humanized monoclonal anti-CD33 IgG4 (immunoglobulin G4) antibody conjugated to the payload calicheamicin at an average drug-antibody ratio of 2 to 3. The unarmed antibody has been shown to have no cytotoxic potency against acute myeloid leukemia (AML) cells in vitro and to mediate neither complement-binding nor antibody-dependent cell-mediated cytotoxicity (ADCC) in vivo [4]. The bifunctional 4-(4-acetylphenoxy) butanoic acid (AcBut) linker of GO attaches on the one end via amid bonds to surface-exposed lysine residues of the antibody and forms on the other end an acyl hydrazone linkage with the payload. Upon internalization of GO, acidification in the lysosomal pathway releases calicheamicin as a prodrug that is activated by undergoing spontaneous reaction with reduced glutathione within the cytosol [5]. The CD33 surface antigen is an inhibitory receptor and adhesion molecule, whose normal expression pattern is largely restricted to cells from the myeloid lineage, including Kupffer cells and circulating macrophages [6]. CD33-positive leukemia is defined as either presence of CD33 on greater than 20–25% of the leukemic blasts or by CD33 immunofluorescence staining greater than fourfold above background or by 80% CD33-positive cells by flow cytometry [7]. Despite the heterogeneity of CD33 expression on AML cells, more than 80% of patients fulfill at least one of these criteria for CD33-positive disease. In May 2000, Gemtuzumab ozogamicin (GO) received accelerated FDA approval as a stand-alone treatment for CD33-positive AML patients of 60 years or older in first relapse who are not candidates for other chemotherapies. However, Pfizer voluntarily withdrew the ADC from the U.S. market in 2010, after subsequent trials failed to verify its clinical benefit and, even worse, observed a high number of early deaths due to veno-occlusive liver disease. In 2014, a meta-analysis of five randomized trials spurred renewed interest in the therapeutic potential of GO by showing that 5-year survival rates for patients with favorable cytogenetics and intermediate risk were improved by adding GO to induction chemotherapy [8]. In follow-up studies, GO at reduced doses was confirmed to improve survival compared to best supportive care with a favorable risk/benefit profile [9]. In September 2017, the FDA then re-approved GO at a 3-fold lower recommended dose and with a different dosing schedule as either monotherapy or in combination with chemotherapy for the treatment of adults with newly diagnosed CD33-positive AML as well as for the treatment of relapsed/refractory AML patients.

Currently, GO is the only FDA-approved, CD33-targeted therapy, since clinical development of the next most advanced CD33-targeted ADC, which used a PBD payload, was stopped due to increased rates of death in the treatment versus control arm [10]. Although CD33 surface levels are a key determinant of how much GO can bind to AML cells, assessing whether patients with higher CD33 expression levels benefit more from GO therapy is complicated by the fact that CD33 levels are associated with established prognostic factors, including genetic subgroups. However, a recent study demonstrated CD33-positivity to independently correlate with GO benefit for younger and older adults with AML except for AML forms characterized by cytogenetic rearrangements that disrupt genes encoding subunits of a transcription factor known as a core-binding factor [11]. Another recent study determined the impact of single nucleotide polymorphism (SNP) in the splice enhancer region of the CD33 gene that regulates the expression of an alternatively spliced CD33 isoform without exon 2, which contains the antibody binding site for GO [12]. Patients with the CC genotype of this SNP had significantly lower levels of the CD33 isoform lacking exon 2 and, in contrast to the other SNP genotypes, their risk of relapse was almost halved by adding two doses of GO to a standard five-course chemotherapy regimen.

Acquired resistance to GO was frequently observed to be mediated by multidrug resistance mechanisms [13]. In vitro treatment with inhibitors of p-glycoprotein (MDR1/multidrug resistance protein 1/ABCB1) and other multidrug resistance proteins increased cytotoxic potency of GO on resistant cell lines and patient samples [14,15]. In HL-60 cells made GO resistant by chronic exposure, it was shown that MDR-1 is highly expressed. While MDR-1 up-regulation was reversible upon GO withdrawal, the resistant cells, in contrast to the parental line, retained the ability to rapidly re-induce MDR-1 upon re-exposure [16]. Activated signaling via the phosphatidylinositol-3-kinase (PI3K)/AKT pathway is another mechanism associated with GO resistance in vitro in primary AML cells. MK-2206, a selective AKT inhibitor, significantly sensitized resistant human AML cells to GO and free calicheamicin [17]. These findings support the notion that DNA damage is necessary but not sufficient for cell killing by ADCs with a calicheamicin payload since the ability of cells to repair DNA damage and to activate downstream anti-apoptotic factors can critically modulate GO cytotoxicity. The mechanism underlying PI3K/AKT-mediated GO resistance appears to increase survival signaling via activation of anti-apoptotic factors. In agreement with this, GO treatment was found to induce p38 stress kinase activation and to cause the cell death mediators, Bak and Bax, to adopt a pro-apoptotic conformation in two sensitive AML cell lines but not in a resistant one [18]. Maimaitili et al. found that in six of eight tested AML cell lines, the cytotoxicity of GO could be synergistically enhanced by concurrent treatment with the mTORC1/2 inhibitor, PP242 [19]. GO sensitization of cells by PP242 is mediated by a dual mechanism that combines enhanced lysosomal function with blockage of activation of the checkpoint kinase Chk1, a key regulator of DNA damage-induced cell cycle arrest.

### 2.2. Brentuximab Vedotin

In 2011, Brentuximab vedotin (BV, Adcetris™) was FDA-approved for the treatment of two blood cancer indications: in Hodgkin’s lymphoma (HL) for patients that have failed an autologous stem cell transplant (ASCT) and in systemic anaplastic large cell lymphoma (sALCL) for patients that have relapsed after multi-agent chemotherapy. BV consists of SGN-30, a chimeric IgG1 antibody against CD30, conjugated by a protease-cleavable valine-citrulline peptide linker to the membrane permeable tubulin polymerization inhibitor, monomethyl auristatin E (MMAE). CD30, as a surface marker, is normally restricted to activated T- and B-cells and is implicated in autoimmune regulation, but on certain lymphoid cancers, its surface levels are substantially elevated. While BV treatment has generated compelling response rates in both indications, treatment with unarmed SGN-30 at up to 7-fold higher doses than BV yielded no objective responses in HL and only a 17% response rate in ALCL [20]. This suggests that the antibody component of BV does not significantly contribute to overall efficacy *per se* neither by functionally blocking CD30 nor by recruiting immune effector cells. Long-term follow-up of the pivotal study that led to approval of BV treatment for HL patients demonstrated 5-year rates of 22% for progression-free survival (PFS) and of 41% for overall survival (OS) [21]. Thirty-four percent of treated patients initially achieved a complete response indicating that their leukemic cells were not innately resistant to BV. The 5-year outcome of such complete responders was even more favorable with PFS and OS rates of 52% and 64%, respectively. But still, the fact that more than a third of the complete responders relapsed and died within 5 years of treatment shows that their leukemic cells acquired resistance to BV.

Chen et al. investigated resistance mechanisms in vitro by constant BV exposure of an ALCL cell line and pulsatile BV exposure of an HL cell line [22]. The cell lines escaped from continued or repeated cytotoxic effects of BV by various mechanisms that included down-regulating CD30, up-regulating the p-glycoprotein drug transporter, and developing resistance to MMAE. While up-regulation of members of the drug transporter class was also confirmed by immunohistochemistry in clinical samples from BV resistant HL and ALCL patients, none of these samples showed down-regulation of CD30. Moreover, in vitro resistance was neither correlated with the percentage of CD30-positive cells nor with the median cell surface CD30 signal intensity for a given cell population. Thus, it appears that factors other than regulation of CD30 levels contribute substantially to the resistance phenotype. Drug transporters, like p-glycoprotein, preferentially export hydrophobic cargo out of cells. Therefore, one way to overcome the effect of their up-regulation would be to replace the non-charged MMAE payload and peptide linker of BV with a payload/linker combination that produces charged or strongly polar metabolites [23]. The caveat of such a strategy is that it improves potency on multidrug-resistant tumor cells at the expense of limiting the so-called bystander effect. Bystander killing of surrounding tumor cells can occur when active metabolites of ADCs are either prematurely released outside of cells or diffuse out of tumor cells after ADC internalization and degradation or leak from dying tumor target cells. The ability of ADC payload metabolites to cross the biomembranes of surrounding cells determines the extent of bystander killing and this ability is decreased for charged or polar molecules.

While preclinical and clinical findings show that leukemic cells can adapt to the cytotoxic stress exerted by BV treatment, such resistance-mediating adaptations need not necessarily be stable after treatment stops. A Phase II study investigated BV retreatment of 20 HL patients and eight sALCL patients who previously had achieved complete or partial response with this drug. For HL patients, retreatment yielded an objective response rate of 60% and a complete response rate of 30%. For sALCL patients, response rates were even better with 88% and 63%, respectively [24]. The median duration of renewed partial or complete responses was 9.2 and 12.3 months for the two patient populations, respectively. This demonstrates that retreatment after therapy discontinuation due to emerging acquired resistance is a therapeutic option for many patients. However, it should be noted that peripheral neuropathy is a known adverse effect of BV treatment that occurred more frequently in retreated patients (~30%). In fact, most patients discontinuing retreatment did so for this reason. Peripheral neuropathy must be clinically managed by dose reduction potentially limiting the efficacy of BV in these patients. The instability of the BV resistance phenotype after treatment discontinuation suggests that underlying stress adaptations come at a certain cost for the vitality of leukemic cells.

### 2.3. Trastuzumab Emtansine

In 2013, the FDA approved Trastuzumab emtansine (T-DM1, Kadcyla™) for the treatment of HER2-positive metastatic breast cancer. T-DM1 consists of the humanized anti-HER2 IgG1 antibody trastuzumab conjugated via a non-cleavable succinimidyl 4-(N-maleimidomethyl) cyclohexane-1-carboxylate (SMCC) linker to DM-1, a microtubule inhibiting maytansinoid. Two properties set T-DM1 apart from most other ADCs: (i) due to gene amplification, its target antigen HER2 is extremely abundant on the targeted tumors [25] with more than one million molecules per cell (this is about 10–100 fold higher than other ADC targets) and (ii) trastuzumab, as an unarmed antibody (Herceptin™), was already FDA-approved for the same indication in 1998 [26], based on its ability to functionally neutralize HER2 and to also exert antibody-dependent cell-mediated cytotoxicity (ADCC). As monotherapy, trastuzumab is given at a loading dose of 8 mg/kg in the first week and from week 4 onwards, at 6 mg/kg every 3 weeks, while T-DM1 is dosed continuously at 3.6 mg/kg every 3 weeks from start. Thus, despite somewhat lower antibody exposure for T-DM1 than for trastuzumab treatment, functional blocking of HER2 signaling and ADCC could conceivably contribute to the overall efficacy of T-DM1.

Two Phase III trials established T-DM1 as a standard of care in second or higher line therapy for HER2-positive, metastatic breast cancer patients. In the EMILIA trial [27], T-DM1 was compared to oral combination therapy with capecitabine, a 5-fluorouracil prodrug, plus the HER2 inhibitor, lapatinib. It improved PFS by 3.2 months and OS by 4 months, while almost doubling the duration of response to 12.5 months. The TH3RESA trial compared T-DM1 versus treatment of physician’s choice in locally advanced or metastatic breast cancer patients who had received a taxane in any setting and had progressed after treatment with two or more HER2-directed regimens in the advanced setting [28,29]. In this difficult to treat patient population, T-DM1 almost doubled PFS and increased OS by 7 months despite the fact that about half of the patients in the physician’s choice group were allowed to cross over to T-DM1 therapy. The fact that all treated patients had previously progressed on a microtubule inhibiting drug, as well as on at least two anti-HER2 regimens, clearly illustrates that the efficacy of an ADC is more than just the sum of its parts. Importantly, T-DM1 was also better tolerated than the control treatments in all Phase III trials.

The EMILIA trial also reported an interesting difference in therapeutic outcome between treatment arms for patients with PIK3CA mutations [30]. Cancer cells with such mutations become resistant to HER2 signaling blockade due to constitutive activation of the alternative signaling pathway triggered by PI3K. While PFS was similar for T-DM1-treated patients with and without PIK3CA mutations (10.9 vs. 9.8 months, respectively), it was worse for patients treated with lapatinib and capecitabine (4.3 vs. 6.4 months). This indicates that for the overall clinical efficacy of T-DM1, the targeting function of trastuzumab might play a bigger role than its HER2 signaling inhibition functionality.

More recently, the Phase III MARIANNE study investigated T-DM1 as first-line HER2 therapy in patients with progressive/recurrent locally advanced breast cancer or untreated metastatic breast cancer. With regards to PFS as the primary endpoint, T-DM1 monotherapy, as well as its combination with pertuzumab, another FDA-approved HER-2 antibody, was non-inferior but also not superior to combination therapy with trastuzumab plus a taxane [31]. Although the reasons for this are not fully understood, it should be noted that T-DM1 has a positively charged payload and hence there is no bystander effect, whereas taxanes readily cross biomembranes and are therefore also toxic for tumor cells with low or no HER2 expression. Under selective pressure by anti-HER2 treatment, subclones with limited or no HER2 expression can become dominant in a tumor that has ab initio heterogeneous HER2 expression [32]. This will gradually result in increased resistance to T-DM1 but not to taxanes [33] and adding pertuzumab to T-DM1 does not change this. Perhaps not too surprising then, the phase III KRISTINE trial [34] found that for neoadjuvant treatment of HER2-positive patients with localized, operable breast cancer combining T-DM1 with pertuzumab was inferior to a combination of trastuzumab, pertuzumab, and two chemotherapeutics (docetaxel and carboplatin) with regards to pathologic complete response. However, T-DM1 was superior to trastuzumab in the recently published phase III KATHARINE trial comparing the adjuvant treatment of patients with HER2-positive early breast cancer who had residual invasive disease at surgery following neoadjuvant therapy containing a taxane and trastuzumab. After random assignment to 14 cycles of adjuvant therapy, the risk of recurrence of invasive breast cancer or death was cut in half by T-DM1 versus trastuzumab treatment [35]. Given the clinical success of T-DM1, it’s not surprising that there are currently at least nine other HER2-based ADCs in clinical development [36].

Due to its lack of bystander effect, T-DM1 tumor uptake and therapeutic response are expected to correlate closely with expression levels of HER2. Preclinical confirmation comes from microPET/CT imaging of zirconium-78 labeled T-DM1 in mouse xenograft studies [37], while clinical proof is provided among others by a biomarker analysis of the TH3RESA trial demonstrating that patients with higher HER2 mRNA levels benefited more from receiving T-DM1 [38]. In agreement with this, Loganzo et al. found by exposing HER2-overexpressing cell lines in vitro to multiple cycles of T-DM1 treatment at IC_80_ levels that HER2 down-regulation was the primary mechanism of acquired resistance besides the up-regulation of drug efflux transporters [39]. In another preclinical study, Sung et al. also reported down-regulation of HER2 as a mechanism of acquired resistance in two breast carcinoma cell lines, while a gastric carcinoma cell line developed T-DM1 resistance by altered internalization and trafficking of T-DM1 instead [40]. In the latter cell line, proteins that mediate caveolae formation and endocytosis were enriched, and T-DM1 was internalized into intracellular caveolin-1 positive endocytic compartments with neutral pH. These endosomes differ with regards to pH and proteinase activities from the increasingly acidic endo-lysosomal maturation pathway, for which T-DM1 is destined in sensitive cells. Not surprisingly, the same cell line was cross-resistant to trastuzumab-ADCs comprising auristatin analogs conjugated by a non–cleavable maleimide linker because of its inability to fully degrade the antibody moiety in endosomes which prevents payload release. Resistance was overcome, however, when the same ADCs contained a protease-cleavable valine-citrulline peptide linker instead. The payloads could then be released by linker cleavage and, unlike lysine-MCC-DM1, the active metabolite of T-DM1, the uncharged auristatin analogs are able to escape into the cytosol by crossing the endocytic membrane.

Diminished cytosolic accumulation of lysine-MCC-DM1 was also observed in another highly T-DM1 resistant gastric carcinoma cell line derivative [41]. In that case, reduced activity of the vacuolar H+-ATPase (V-ATPase) led to diminished lysosome acidification, which, in turn, affected lysosomal proteinase activities and prevented efficient degradation of T-DM1. In an independent study, Rios-Luci et al. also reported that resistant cell clones with increased lysosomal pH had impaired proteolytic degradation of T-DM1 [42]. Moreover, Hamblett et al. discovered that the active metabolite, lysine-MCC-DM1, requires the presence and activity of the lysosomal transporter, SLC46A3, in order to escape into the cytosol and that down-regulation of SLC46A3 can render cells T-DM1 resistant [43]. A combined loss of SLC46A3 and PTEN functionality was reported to contribute to T-DM1 resistance in yet another breast cancer cell line [44]. Silencing of SLC46A3 conferred partial resistance, and a PI3K inhibitor sensitized cells to T-DM1. Thus, the resistance observed in the reported cell line is likely a combined effect of the reduced cytosolic escape of lysine-MCC-DM1 from lysosomes and apoptosis resistance due to constitutive survival signaling by the PI3K pathway. Wang et al. reported, for a breast cancer cell line, another anti-apoptotic signaling mechanism resulting in T-DM1 resistance in vitro. They found that aberrant activation of STAT3 due to overexpression and ligand-induced signaling of the LIF (Leukemia Inhibitory Factor) receptor blocked apoptosis induction by T-DM1 [45].

Moreover, PLK1, the mitotic Polo-like kinase 1, mediates T-DM1 resistance by overriding spindle assembly checkpoint-dependent mitotic arrest [46]. Genomic or pharmacological inhibition of PLK1 restored T-DM1 sensitivity also by cdk1-dependent phosphorylation and by inactivation of the apoptosis guards Bcl-2/Bcl-xL. Cyclin B1, which together with cdk1 forms the so-called maturation promoting factor responsible for the switch-like commitment of cells to mitosis, has also been implicated in T-DM1 resistance. In fresh HER2-positive human breast tumor explants, induction of cyclin B1 by T-DM1 was reported to correlate with apoptosis induction. Also, in breast cancer cell lines made resistant to T-DM1 in vitro, the ADC failed to induce cyclin B1 [47]. Finally, while in vitro tubulin gene mutations [48] and down-regulation of βIII and βV-tubulins [49] in tumor cells have been shown to play a role in acquired resistance to taxanes, there is currently no preclinical or clinical evidence for their relevance for ADC resistance. Nevertheless, it has become clear that under T-DM1-induced cellular stress, cancer cells can develop a multitude of resistance phenotypes by reducing the effectiveness of various steps in the cascade of events that lies between ADC uptake and cell death induction.

### 2.4. Inotuzumab Ozogamicin

In August 2017, Inotuzumab ozogamicin (INO, Besponsa™) gained FDA approval for the treatment of adults with relapsed or refractory B-cell precursor acute lymphoblastic leukemia (ALL). INO uses an IgG4 antibody directed against the B-lymphocyte cell adhesion molecule CD22 as the targeting moiety and a DNA double-strand break inducing calicheamicin analog as payload. As with GO, an acid-labile hydrazone linkage releases the payload in late endosomes/lysosomes from the antibody moiety. Over 90% of ALL patients express CD22 on leukemic blasts [50,51]. CD22 is constitutively endocytosed and degraded [52], which makes it an ideal target for payload delivery in general. Currently, there are four more CD22-based ADCs in various stages of preclinical and clinical development for different blood cancer indications [53,54,55,56]. The first FDA-approved immunotoxin, Lumoxiti™ from Astra-Zeneca, which achieved 75% objective response rateand 30% durable complete responses in hairy cell leukemia, also targets CD22.

In the phase 3 INO-VATE ALL trial, treatment with INO outperformed standard intensive chemotherapy of ALL patients with regards to complete remission rate (80.7% versus 29.4%, respectively) and OS (median 7.7 versus 6.7 months) as endpoints [57]. Veno-occlusive liver disease occurred as a serious adverse effect of INO in 11% of treated patients. Although clinical responses with INO therapy have not been durable, there are otherwise limited treatment option for this aggressive disease, and the time gained may be enough to bridge patients to stem cell transplants [7]. So far, resistance to INO has been primarily reported to occur by the up-regulation of drug transporters that efflux the released payload from the cytosol before it can translocate to the nucleus and damage DNA. In cell lines and primary patient samples, the cytotoxic potency of INO was inversely correlated to the expression levels of P-glycoprotein [58].

## 3. Challenges to Overcome ADC Resistance Mechanisms

As outlined in the text above and illustrated in Figure 1, the complex mode of action of ADCs offers tumors a plethora of escape mechanisms. In targeted drug development for oncology, next-generation drugs are typically designed based on an improved understanding of target biology and of tumor escape mechanisms, as has been done with second and third generation BCR/ABL tyrosine kinase inhibitors in chronic myeloid leukemia. Following the same thinking, it has been proposed to exploit the modular nature of ADCs by swapping individual components of the molecule with others having different functional properties to overcome resistance [59]. For example, enabling bystander killing by exchanging the linker-payload elements of T-DM1 for a cleavable linker with a membrane permeable payload has proven successful in some preclinical resistance models [39]. However, such modified trastuzumab-based ADCs have only been tested in a preclinical setting, and it is unclear, how much cardiotoxicity they might cause in patients. The reason for choosing the particular linker/payload combination that is used in T-DM1 was to avoid on-target toxicity due to known HER2 expression on cardiomyocytes, and indeed no significant cardiotoxicity has been observed in the clinic with T-DM1, even in heavily pre-treated patients [60].

In general, the proposed “component switch” strategy will be severely limited by the fact that exchanging functional components will inevitably have difficulty to predict consequences on the anyways narrow therapeutic index of an ADC. Moreover, instead of blocking only a single event, like payload release, tumor cells will likely adapt under continued selective pressure by slightly modifying several of the various steps that have to occur in succession for an ADC to exert its cytotoxic effect such that the net effect of these changes is a strong resistance phenotype. Hence, in principle, there could be a wide range of different resistance phenotypes, most of which would need to be addressed differently. Last but not least, the ability to assess the contribution of various resistance mechanisms in clinical samples is severely limited not only by current lack of sequential pre- and post-treatment biopsies but also by the technical challenge of establishing validated, quantifiable clinical assays for protein levels of resistance biomarkers.

A conceptually intriguing approach to increase the efficiency of payload delivery to the cytosol, and hence the cytotoxic potency of ADCs, has recently been patented by Ward and coworkers from Texas University [61]. They engineered the targeting moiety of an ADC such that the binding affinity for its target is by two orders of magnitude lower in the endolysosomal milieu (pH < 6.5; [Ca^2+^] ~2 µM) than in the extracellular space (pH > 6.8; [Ca^2+^] ~2 mM). The net result is early dissociation of the ADC from its target, which can then be recycled to the cell surface and capture and internalize more ADC molecules. This could be a smart way to curb resistance mechanisms that diminish cytosolic escape of the payload since it only enhances target-mediated but not target-independent payload uptake and hence should also increase the therapeutic window.

## 4. Promising Combination Therapy Approaches

Purposely selected combination therapies seem to be a more promising strategy to overcome or avoid resistance mechanisms. Simultaneous or frequently alternating administration of ADCs with other cytotoxic agents with a different mode of action reduces the likelihood that individual tumor clones become simultaneously resistant to both agents, while in combination therapy with immune-checkpoint inhibitors, the ADC component is intended to enhance anti-tumor immunity by immunogenic cell death and enhanced tumor antigen release.

### 4.1. The Combination with Chemotherapeutics and/or Targeted Agents

For combining an ADC with another cytotoxic agent, the two drugs should not have overlapping adverse effect profiles in order to avoid additive toxicities. This is illustrated by a phase 1b/2a metastatic breast cancer trial that tested combining T-DM1 with docetaxel (with and without pertuzumab). While docetaxel plus T-DM1 appeared efficacious, nearly half of the patients experienced severe adverse effects that required dose reductions [62]. Equally important is that the combined cytotoxic agents need to have unique modes of action in order to prevent cross-resistance. For example, in vivo resistance to pinatuzumab vedotin, an anti-CD22-vc-MMAE ADC, did not endow cross-resistance to an ADC that used a different payload, a highly potent anthracycline analog [56]. However, under selective pressure from simultaneous treatment with both agents, tumor cells might instead evolve different escape mechanisms that exploit aspects shared by both ADCs. In this case, where the two ADCs share the same antibody moiety and linker, tumor cells might escape, for example, by down-regulating CD22 or by changing its internalization kinetics or intracellular trafficking.

Obviously, the less two agents have in common, the less likely cross-resistance will occur. A recombinant immunotoxin consisting of the anti-HER2 diabody C6.5 fused to a de-immunized form of the plant toxin bouganin has been shown to overcome cross-resistance to both T-DM1 and a trastuzumab-MMAE conjugate due to its completely different mode of action [63]. Although the auristatin-based ADCs, as well as the immunotoxin both target HER2, the epitope of the C6.5 diabody does not overlap with that of trastuzumab, thereby allowing simultaneous treatment without competition for target binding. This is important, since, according to the Goldie-Coldman hypothesis, the evolution of acquired resistance is best prevented by early treatment start and simultaneous combination therapy with multiple drugs. In general, immunotoxins are ideal combination agents especially for anti-microtubule agent-based ADCs because (i) their intracellular processing differs from that of ADCs; (ii) they have a highly potent, unique mode of action (i.e., protein synthesis inhibition); (iii) their adverse effect profile does not overlap with microtubules-inhibiting ADCs; (iv) protein toxins are no substrates for MDR transporters and (v) immunotoxins have been shown to act synergistically with taxanes [64].

Another drug class with a unique mode of action are antibody targeted amanitin conjugates (ATACs), ADCs with α-amanitin as payload. By studying drug-tolerant cancer cell colonies that survive in the presence of different cytotoxic drugs, Kume et al. found that the RNA-polymerase II inhibitor, α-amanitin, consistently and potently inhibited the drug-tolerant phenotype [2]. Mechanistically, this was mediated by suppressed expression of *TAF15*, a gene encoding an RNA-binding factor that regulates transcription and RNA processing. Once ATACs have proven their clinical utility, their ability to broadly inhibit resistance against other drugs could make them promising combination agents for other ADCs. However, the first ATAC from Heidelberg Pharma is expected to enter the clinic only later this year.

Clinical evaluation of ADC combinations is ongoing for a multitude of targeted agents and chemotherapeutics that are either already approved or in late-stage clinical development. For example, there is great interest in extending the clinical success of CDK4/6 inhibitors in hormone receptor-positive luminal breast cancer to metastatic HER2+ breast cancer by combination therapy with T-DM1. Table 2 summarizes such promising ongoing clinical trials that test FDA-approved ADCs in combination with standard chemotherapeutics and targeted small molecule inhibitors.

### 4.2. Combination of ADCs with Immune Checkpoint Blockade

Immune checkpoint inhibiting (ICI) antibodies have revolutionized cancer therapy by achieving long-lasting responses and even cures in subsets of patients. They do so by enhancing anti-tumor immune responses of effector T-cells and stimulating anti-tumor immunological memory in patients. ICI antibody therapy works well for “inflamed” (hot) tumors that have already an ongoing anti-tumor T-cell response, but not for “immune desert” or “immune excluded” (cold) tumors, where there is no anti-tumor T-cell response or T-cells are excluded from the tumor, respectively [65]. Chemotherapeutics and ADCs can principally enhance anti-tumor immunity by three distinct mechanisms (reviewed in [66]), (i) release of tumor antigens from dying cancer cells; (ii) direct effects of the free payload on maturation and activation of antigen-presenting dendritic cells; (iii) induction of immunogenic cell death of tumor cells. Immunogenic cell death is characterized by the release of damage-associated molecular patterns (DAMPs) that act as danger signals by stimulating specific receptors on dendritic cells.

Combining ADCs with ICI antibody therapy could increase the percentage of patients with complete and durable responses by turning immunologically cold tumors into hot ones using the three mechanisms outlined above. Looking at it the other way round, the key attribute that makes ICI antibodies attractive as combination agents for enhancing ADC therapy is that a strong anti-tumor immune response leads to antigen-cross-presentation, which then allows the immune system to destroy tumor cells irrespective of whether they have down-regulated or completely lost the surface antigen against which the ADC is directed. In a HER2-positive orthotopic tumor model that was inherently resistant to treatment with a combination of ICI antibodies, co-treatment with T-DM1 led to tumor rejections accompanied by massive infiltration of T-cells into tumors [67]. The same publication also reports that T-DM1 increased T-cell infiltration into primary human breast tumors in the WSG-ADAPT trial. Only a limited number of untargeted chemotherapeutics and only BV of the FDA-approved ADCs have been reported to induce immunogenic cell death. However, a wide variety of anti-tubulin agents, including MMAE and DM-1, were shown to directly induce phenotypic and functional maturation and activation of dendritic cells in vitro and in vivo suggesting that this is a class effect [68]. On the downside, severe, potentially target mediated lymphopenia and neutropenia, a known adverse effect of MMAE and DM1, are of concern for clinically combining ADCs and ICI antibodies, as they could hamper anti-tumor immunity. However, such interference, if it occurred, could potentially be mitigated by sequential rather than simultaneous administration. Ultimately, clinical trials will need to show, whether ADCs can substantially enhance anti-tumor immunity thereby improving response rates and increasing the numbers of long-term survivors in combination therapy with immune checkpoint blockade. Table 3 summarizes the ongoing clinical trials which evaluate the combinations of ADCs with ICI antibodies in different cancer indications.

## 5. Summary

Although few, the currently FDA-approved ADCs provide valuable treatment options for difficult to treat patient populations. The recent establishment of site-specific conjugation methods promises to increase the rather poor success rate for drug development of ADCs to date. Primary and acquired resistance are frequently encountered limitations for this very potent class of agents. Their complex mode of action requires a multi-step cascade of events to occur efficiently for a sufficient amount of toxic payload to be delivered inside the targeted cancer cells. This provides tumor cells with many opportunities to thwart an ADC attack. The most promising avenues for achieving more long-term therapeutic benefits for a larger percentage of patients are deliberately chosen combination therapies.

## Figures and Tables

**Figure 1 cancers-11-00394-f001:**
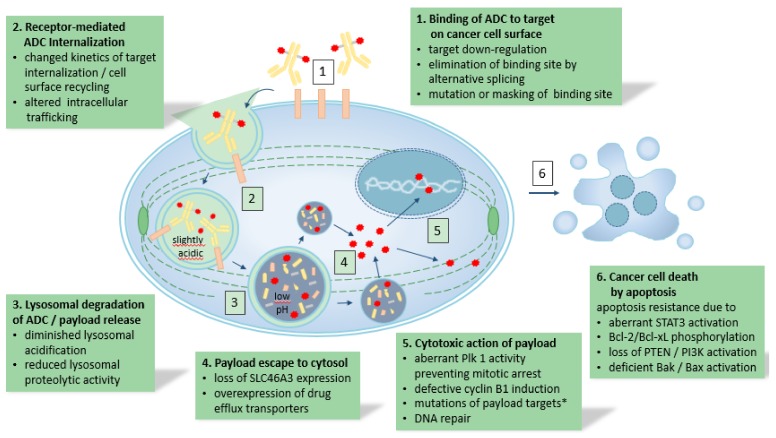
The sequence of events that have to occur for an ADC to exert its cytotoxic action and mechanisms of resistance that can affect them. * such mutations are known to play a role in resistance to taxanes, but the same has not been shown yet for ADCs with microtubule-disrupting payloads.

**Table 1 cancers-11-00394-t001:** Documented resistance mechanisms for FDA-approved antibody-drug conjugates (ADCs).

ADC	Resistance Mechanisms Directed Against		
	Targeting Moiety	Linker	Payload
Gemtuzumab ozogamicin	CD33 splice variant lacking antibody epitope		drug efflux transporters,
			PI3K pathway activation,
			mTORC1/2 activation,
			deficient Bak/Bax activation
Brentuximab vedotin	CD30 down-regulation		drug efflux transporter,
			MMAE resistance
Trastuzumab emtansine	HER2 down-regulation,	enhanced trafficking to non-lysosomal compartments, reduced V-ATPase activity *	drug efflux transporter,
			SLC46A3 down-regulation,
			STAT3 pathway activation,
	altered internalization		PTEN/PI3K activation,
			PLK1 activation,
			failure to induce Cyclin B1
Inotuzumab ozogamicin			drug efflux transporters

* non-cleavable linker requires complete antibody degradation by lysosomal enzymes. FDA = Food and Drug Administration; CD = Cluster of Differentiation; PI3K = PhosphatidylInositol-3-Kinase; mTORC = mammalian Target Of Rapamycin Complex; Bak = Bcl-2 antagonist/killer; Bax = Bcl-2-associated x protein; MMAE = MonoMethyl Auristatin E; HER2 = Human Epidermal growth factor Receptor 2; SLC46A3 = SoLute Carrier 46A3; STAT3 = Signal Transducer and Activator of Transcription 3; PTEN = Phosphatase and TENsin homolog; V-ATPase = Vacuolar-type proton pumping Adenosine Tri-Phosphate hydrolyzing enzyme; PLK1 = Polo-Like Kinase 1.

**Table 2 cancers-11-00394-t002:** Selected ongoing clinical trials combining antibody-drug conjugates (ADCs) with chemotherapeutics and targeted agents.

Combination	Trial Number	Phase	MoA of Combined Agent(s)	Indication
GO and 5-azacitidine	NCT00766116	I/II	DNA methyl-transferase inhibitor	Relapsed AML
GO and the combo of idarubicin, etoposide, cytarabine, pegfilgrastim with ATRA ^1^	NCT00893399	III	Three different DNA damaging agents plus agonists for G-CSF and retinoic acid receptor	AML with NPM1 mutation ^2^
GO and glasdegib	NCT03390296	Ib/II	Smoothened inhibitor	RR ^3^ AML
Conditioning therapy with GO plus cyclophosphamide and busulfan chemotherapy followed by ASCT	NCT02221310	II	Two immune suppressive alkylating agents	High-risk AML or myelodysplastic syndrome (MDS)
GO and G-CSF, cladribine, cytarabine, and mitoxantrone	NCT03531918	I/II	Three different DNA damaging agents plus a G-CSFR agonist	1st line AML
GO and daunorubicin/cytarabine filled liposomes	NCT03672539	I	Two different DNA damaging agents	RR ^3^ AML, high-risk MDS
Neoadjuvant T-DM1 and lapatinib followed by Abraxane	NCT02073487	II	A HER1/2 kinase and a microtubule inhibitor	HER2^+^ breast cancer
T-DM1 and poziotinib	NCT03429101	I	Covalent HER1/2/4 kinase inhibitor	Metastatic HER2^+^ breast cancer
T-DM1 and osimertinib	NCT03784599	II	HER1 T790M kinase inhibitor	Mutant HER1^+^, HER2^+^, stage IV lung cancer
T-DM1 and neratinib	NCT02236000	Ib/II	Irreversible pan-HER inhibitor	Metastatic HER2^+^ breast cancer
T-DM1 and palbociclib	NCT01976169	Ib	CDK4/6 inhibitor	Recurrent or metastatic HER2+ breast cancer
T-DM1 and ribociclib	NCT02657343	Ib/II	CDK4/6 inhibitor	Metastatic HER2^+^ breast cancer
T-DM1 and palbociclib	NCT03530696	II	CDK4/6 inhibitor	Metastatic HER2^+^ breast cancer
T-DM1 and taselisib	NCT02390427	Ib	Phosphoinositide 3-kinase α inhibitor	Advanced HER2^+^ breast cancer
INO and cyclophosphamide, vincristine, prednisone	NCT01925131	I	An alkylating agent, a microtubule inhibitor, and a glucocorticoid	RR ^3^ CD22^+^ acute leukemia
INO and low dose chemotherapy (cyclophosphamide/vincristine or methotrexate/cytarabine)	NCT01371630	I/II	Well tolerated cytostatic agents	Older patients with previously untreated ALL
INO and bosutinib	NCT02311998	I/II	Bcr-Abl kinase inhibitor	CD22 and Philadelphia-chromosome positive ALL and CML
INO and intensive chemo-therapy (Hyper-CVAD regimen)	NCT03488225	II	11 induction therapy plus two maintenance therapy agents	1st line B-cell lineage ALL
INO and rituximab, cyclophosphamide, vincristine, prednisolone	NCT01679119	II	Anti-CD20 antibody, well tolerated cytostatic agents	DLBCL patients unfit for anthracycline

GO = Gemtuzumab ozogamicin (Mylotarg ™), T-DM1 = trastuzumab emtansine (Kadcyla ™), INO = Inotuzumab ozogamicin (Besponsa ™), MoA = Mechanism of Action, G-CSF/G-CSFR = Granulocyte–Colony-Stimulating Factor and its Receptor, AML = Acute Myeloid Leukemia, ASCT = Autologous Stem Cell Transplant, ALL = Acute Lymphoblastic Leukemia, CML = Chronic Myeloid Leukemia, DLBCL = Diffuse Large B-Cell Lymphoma, ^1^ all-*trans* retinoic acid, ^2^ nucleophosmin-1, ^3^ relapsed/refractory.

**Table 3 cancers-11-00394-t003:** Selected ongoing clinical trials which have combined antibody-drug conjugates (ADCs) with immune checkpoint inhibitor agents.

Combination	Trial Number	Phase	MoA of Combo Agent(s)	Indication
T-DM1 and pembrolizumab	NCT03032107	Ib	PD-1 blocking antibody	Metastatic HER2^+^ breast cancer
T-DM1 and atezolizumab	NCT02924883	Ib	PD-L1 blocking antibody	Locally advanced or metastatic HER2^+^ breast cancer
Different doses of T-DM1 and atezolizumab	NCT02605915	Ib	PD-L1 blocking antibody	Locally advanced or metastatic HER2^+^ breast cancer
T-DM1 and utomilumab	NCT03364348	Ib	Agonistic 4-1BB antibody	HER2^+^ advanced breast cancer
BV and nivolumab	NCT02581631	I/II	PD-1 blocking antibody	RR ^1^ CD30^+^ Hodgkin Lymphoma
BV and pembrolizumab	NCT02684292	III	PD-1 blocking antibody	RR ^1^ CD30^+^ Hodgkin Lymphoma
BV and nivolumab +/− ipilimumab	NCT01896999	I/II	PD-1 and CTLA-4 blocking antibodies	RR ^1^ CD30^+^ Hodgkin Lymphoma

T-DM1 = trastuzumab emtansine (Kadcyla ™), BV = Brentuximab vedotin (Adcetris ™), MoA = Mechanism of Action, PD-L1 = Programmed cell Death-Ligand 1, CTLA-4 = Cytotoxic T-Lymphocyte-Associated Protein 4 ^1^ relapsed/refractory.

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
