# Peer review of "Acquired Resistance to Antibody-Drug Conjugates"

_cancers, 2019, doi:10.3390/cancers11030394_

Round 1

Reviewer 1 Report

This is a well written paper which provides a comprehensive review of currently approved antibody-drug conjugates in hematological and solid tumours, with particular emphasis on mechanism of resistance encountered with this class of drugs.

The following issues should be addressed in the manuscript:

- Pg 2, Line 46: "...depending of the chemical..." needs to be changed to "... depending on the chemical..."

- a reference should be provided to support the statement on pg 2, line 92-94: "The unarmed antibody has been shown to have no cytotoxic potency against acute myeloid leukemia 93 (AML) cells in-vitro and to mediate neither complement-binding nor ADCC in-vivo."

- please reference the EMILIA trial pg 5, line 223

- please reference the KRISTINE trial pg 6, line 254

- a reference should be added to the statement on pg 7, line 344- 346: " For example, enabling bystander 344 killing by exchanging the linker-payload elements of T-DM1 for a cleavable linker with a membrane 345 permeable payload has proven successful in some preclinical resistance models."

- I would suggest changing the title of section 3 - to Challenges to overcome ADC resistance mechanisms. The authors have highlighted the difficulties faced with future ADC development, and not proposed/reviewed any options to overcome these in any detail.

- please define ATACs - on pg 9, line 398

Author Response

Thanks for the helpful comments!

We made all suggested edits and added all the requested additional references.

We changed the title of section 3 as proposed, but also made a separate section 4 for the part that talks about promising combination therapies, since we felt that the argument does not apply to this subsection and it would not fit with the new title.

Reviewer 2 Report

In this work, Collins et.al have reviewed the acquired resistance to antibody-drug conjugates which represents a major issue for ADC development. The discussions focus on the approved four ADCs and the whole paper is well-written.

However, there are several revisions needed before publication.

(1)  The authors pointed to combination therapy in the end as the most promising way to overcome resistance, there are also reviews published in recent years claiming the same conclusion. In fact, I think it would be beneficial if the authors can add more on other potential methods (perspective) to inspire the ADC field.

(2)  I suggest the authors to make a table to compare resistance MOA for target, linker and payloads e.g. DNA damaging vs anti-microtubule. Since there are only four ADCs, I think a table will save time for readers and it would not complicate things.

(3)  The authors should also mention the newly discovered ADCs, what are the potential resistance? Any ADC failed due to acquired resistance? I think it will increase the impact of this review and this special issue in Cancers.

Author Response

Thanks for the insightful comments.

Ad 1) We added a brief description of a very recently patented approach for enhancing payload delivery that was brought to our attention by reviewer 3 and that will certainly inspire the field (according to the inventors the corresponding paper is under review at Nature Biotechnology)

Ad 2) We made the proposed additional table and inserted it at the beginning of section 2

Ad 3) Very little is known about resistance to the newly discovered ADCs. So far all terminated ADC programs have failed for lack of therapeutic window, none due to acquired or innate resistance (to the best of our knowledge). Also since the review is already very close to the 6000 word limit, we could not include another section.

Reviewer 3 Report

The manuscript entitled ‘ Acquired resistance to antibody-drug conjugates’ summarized the FDA-approved ADCs and the mechanisms of drug resistance from pre-clinical or clinical studies. The authors also discussed available options to overcome drug resistance.

Generally, this is a very decent review regarding ADCs. It will be very useful for people who are interested in or developing ADCs. There are some grammatical errors, and the authors are supposed to be very careful and revise the whole manuscript again.

To deliver a sufficient amount of toxic payload into target cells or overcome the on-target and off-target toxicities, recently Dr. E.Sally Ward (Texas University) and coworkers improved the potency of antibody-drug conjugates (ADCs) containing the HER2-specific antibody pertuzumab by reducing their affinity by >250-fold for HER2 at acidic endosomal pH relative to near neutral pH. These engineered pertuzumab variants show increased lysosomal delivery and cytotoxicity towards tumor cells expressing intermediate HER2 levels. In HER2 xenograft tumor models in mice, the variants show higher therapeutic efficacy than the parent ADC and a clinically-approved HER2-specific ADC. (patent: WO2018136455A1)

Author Response

Thanks for the constructive comments.

We carefully revised the manuscript and added a short paragraph on the very intriguing work of Dr. Ward and Dr. Ober based on the published patent that you thankfully brought to our attention. Unfortunately the paper is not yet out.